# Receiving Genomic Sequencing Results through the Victorian Undiagnosed Disease Program: Exploring Parental Experiences

**DOI:** 10.3390/jpm12081250

**Published:** 2022-07-29

**Authors:** Jo Martinussen, Michal Chalk, Justine Elliott, Lyndon Gallacher

**Affiliations:** 1Department of Paediatrics, The University of Melbourne, Melbourne 3010, Australia; jo.martinussen@petermac.org (J.M.); michal.faraday@gmail.com (M.C.); 2Victorian Clinical Genetics Services, Murdoch Children’s Research Institute, Melbourne 3052, Australia; justine.elliott@mcri.edu.au

**Keywords:** undiagnosed disease program, genome sequencing, parents, lived experience, genetic counseling, rare disease

## Abstract

Rare diseases cumulatively affect a significant number of people, and for many, a diagnosis remains elusive. The Victorian Undiagnosed Disease Program (UDP-Vic) utilizes deep phenotyping, advanced genomic sequencing and functional studies to diagnose children with rare diseases for which previous clinical testing has been non-diagnostic. Whereas the diagnostic outcomes of undiagnosed disease programs have been well-described, here, we explore how parents experience participation in the UDP-Vic and the impact of receiving both diagnostic and non-diagnostic genomic sequencing results for their children. Semi-structured interviews ranging in length from 25 to 105 min were conducted with 21 parents of children in the program. Ten participants were parents of children who received a diagnosis through the program, and eleven were parents of children who remain undiagnosed. Although the experiences of families varied, five shared themes emerged from the data: (1) searching for a diagnosis, (2) varied impact of receiving a result, (3) feelings of relief and disappointment, (4) seeking connection and (5) moving towards acceptance. The findings demonstrate the shared experience of parents of children with rare disease both before and after a genomic sequencing result. The results have implications for genetic counselors and clinicians offering genomic sequencing and supporting families of children with rare diseases.

## 1. Introduction

Genetic diseases comprise a diverse, often rare range of conditions that can markedly impact the lives of affected individuals and their families. Individually, 80% of genetic conditions are considered rare, although cumulatively, they are thought to affect 6–10% of the population, or more than 2 million Australians, including 400,000 Australian children [1,2,3,4,5,6]. Achieving a genetic diagnosis can involve years of attending medical appointments and undergoing numerous diagnostic tests [7,8,9]. Despite commitment to a lengthy search, often referred to as a “diagnostic odyssey” [10,11,12], many families never receive a formal genetic diagnosis.

The advent of genomic sequencing has brought with it the hope of a transformed diagnostic odyssey for many families. This is attributed to increased diagnostic yield and decreased time for a genetic diagnosis when compared to targeted genetic testing [13,14]. Clinical genomic sequencing may yield diagnostic rates of between 20 and 55%, depending on the primary clinical indication [13,15,16,17]. The high diagnostic rate and the potential for fiscal savings in comparison to traditional technology means genomic sequencing is increasingly recognized as a first-tier diagnostic test for those with a suspected monogenic condition [13,14,18].

Despite increased diagnostic yields, a significant proportion of children who undergo clinical genomic sequencing will not receive a clinical diagnosis. Undiagnosed disease programs (UDPs) have been formed around the world to provide a research pathway for undiagnosed individuals to continue their search for a diagnosis. UDPs aim to harness genomic technology to improve knowledge, management and diagnostic rates of genetic conditions, both in clinical and research settings [7,19,20,21,22]. Possible testing outcomes include not only known genes but also novel gene discoveries. The Victorian Undiagnosed Disease Program (UDP-Vic) commenced in March 2017 and utilizes deep phenotyping, advanced genomic sequencing and functional studies to diagnose children with rare diseases for which previous clinical testing has been non-diagnostic [20]. It is the second such program in Australia.

There is emerging academic literature on the parental experience of receiving a diagnosis for a child through genomic sequencing. Krabbenborg et al. [23] identified three shared hopes of the experience of parents whose children underwent genomic sequencing: information seeking, medical management and finding a supportive environment. Their analysis suggested that although receiving a diagnosis of a rare condition released families from the diagnostic odyssey, it may not provide comprehensive information on the natural history of the disease. Diagnosis of rare or ultra-rare disease is more common through UDPs compared with prior methods of targeted genetic testing, which generally exclude more common conditions [23,24,25]. Rosell et al. [26] reported that following diagnostic testing, a lack of information and the continued inability to connect with families with the same diagnosis may be disempowering for parents. Similarly, parents who received non-diagnostic results were left hoping for more information, despite having low expectations of understanding the cause of the child’s condition [23,26]. Macnamara et al. [27] found that both those who remained undiagnosed or received a new diagnosis had mental health concerns. Their findings emphasized the need to ensure that families have access to ongoing support networks.

The landscape of undiagnosed disease is changing rapidly, with UDPs promising to become a cornerstone of pediatric healthcare combating the diagnostic odyssey through genomic sequencing [21,22] As Australian UDPs are still in their early stages, research on the impact on families and the experience of receiving results through such programs is yet to be comprehensively examined. Understanding this impact is crucial with respect to ensuring that such programs are responsive to the needs of patients and their families.


**Aim**


The aim of the present study was to explore parents’ experiences of receiving either diagnostic or non-diagnostic genomic sequencing results for their child through the UDP-Vic.

## 2. Methods

The study received approval from the Royal Children’s Hospital Human Research Ethics Committee. We adopted a qualitative approach underpinned by a phenomenological framework, with the aims of developing an understanding of how individuals experience a phenomenon in the context of their subjective reality [28,29]. This approach and theoretical framework were chosen to allow participants to share their lived experience of genomic sequencing through the UDP-Vic.

### 2.1. Participant Recruitment

Using purposive sampling, parents/guardians whose children underwent genomic sequencing and received results through the UDP-Vic were recruited for this study. These parents/guardians were identified through the UDP-Vic cohort databases and hospital patient records. They were then assessed for their suitability to participate in this study using the eligibility criteria listed in Table 1. A total of 57 study invitations were sent to eligible parents and guardians to invite them to participate in the study. Depending on family structure, one or both parents/legal guardians were approached to participate. All non-responders were followed-up with a maximum of two phone calls to invite them to participate in the study. Of the remaining invitees, 17 actively declined, and 19 were lost to follow-up. Interviews were organized to take place either via phone or in person, reflecting participant preference.

### 2.2. Data Collection

In-depth, semi-structured interviews were conducted to provide participants a platform to describe their experiences [30]. The interview schedule was developed based upon the research aims and addressed three domains of inquiry: parental experience of child’s undiagnosed condition, participation in genomic investigations and experience of receiving results of genomic investigations (see Appendix A for interview guide). M.C. conducted the interviews with parents whose children received non-diagnostic results from genomic sequencing. J.M. conducted the interviews with parents whose children received diagnostic results. Interviews were audio-recorded, transcribed verbatim and deidentified, and pseudonyms were assigned for analysis. Transcript data were managed using Microsoft Word and Nvivo software.

### 2.3. Data Analysis

Interview transcripts were analyzed using an inductive style of thematic analysis, with the aim of identifying, analyzing and reporting the themes present [31]. To facilitate this process, codes were developed based upon the participants’ own words [29]. These codes were then compared, and similar emergent codes were grouped to form themes [31]. More specifically, we utilized codebook thematic analysis for data analysis by creating a codebook that guided data analysis and theme formation [32]. All transcripts were coded independently by J.M. or M.C. and co-coded by either L.G. or J.E. Coded transcripts were discussed within the research team for rigor.

## 3. Results


**Recruitment results**


In response to the invitation letter and telephone follow-up, twenty-one parents provided consent to participate in interviews. Of these parents, ten were parents of children who had received a diagnostic test result, and eleven were parents of children who had received a non-diagnostic test result. Eighteen mothers and three fathers were interviewed, and interviews ranged in length from 25 to 105 min (Table 2). All participants were offered the option of interviews together with their partner or separately if applicable. All participants opted to be interviewed individually; some participants noted that their partner was unable to attend the interview. All participants and their children were assigned pseudonyms to protect their privacy. Due to the rarity of the conditions and the consequent possibility of identifying the families, we did not name specific genetic diagnoses.


**Thematic Analysis Results**


Within each theme, the spectrum of experiences described by parents of children in both the diagnosed and undiagnosed cohorts is illustrated. The commonalities and unique experiences across both groups of parents are presented (Figure 1).


**Theme 1: Searching for a diagnosis**


Parents across both cohorts shared the experience of entering the UDP-Vic without a formal genetic diagnosis for their child’s presentation. Parents, on average, waited 13 years for a diagnostic result. Parents in the undiagnosed cohort had been searching for a diagnosis for approximately 10 years. Participant descriptions of their motivations for pursuing a diagnosis varied, but their experiences of the process were largely shared. Parents often interchangeably referred to a diagnosis as ‘an answer’, ‘a result’ or ‘a name’.

Most participants described this as an extended process. Following previous failed attempts to diagnose their child, participants described *“not holding their breath”* (Anna, mother of Zoe) about receiving diagnostic results from genomic testing. Although many had low expectations about receiving a diagnosis from genomic testing, participants spoke of feeling *“a glimmer of hope”* (Gina, mother of Riley), believing that this was the most advanced type of test and thus more likely to yield a diagnostic result eventually.


*“We may not have got an answer, we may not have got a diagnosis but obviously with the evolving genetics, it could have been a time factor or down the years that we may have got a diagnosis if we didn’t get one straight away.”*
(Danielle, mother of Paul, diagnostic result)

Some participants discussed the unknown nature of their child’s condition and living without a diagnosis as a significant challenge. This was a motivation for pursuing genetic testing.


*“It’s more scary when it’s uncertain. I was afraid for them and their future of not knowing... I feel like when you don’t know, you’re not in control.”*
(Lana, mother of Maddie, Nelly and Mikey, non-diagnostic result)

Other participants struggled with not knowing what had caused their child’s condition, wondering if they were to blame.


*“I just wanted to know why, what’s the cause of her issues. That had haunted me all along.”*
(Ellie, mother of Tara, non-diagnostic result)

For some participants, the struggle stemmed from uncertainty regarding their child’s future and prognosis.


*“You still didn’t have any actual answers. It was just such a rollercoaster because you’re being told different things. ‘She’s normal’ and then ‘oh no she’s going to die’.”*
(Carly, mother of Winnie, non-diagnostic result)

For others, the pursuit of a diagnosis was motivated by the hope of identifying treatment for their child.


*“I think you’re desperate. I desperately wanted to find something that we could fix...the way to do that was to have a diagnosis.”*
 (Kirsten, mother of Mia, diagnostic result)


**Theme 2: Varied impact of receiving a result**


Participants discussed their perceptions of the UDP-Vic genomic sequencing result with reference to how they had anticipated the result would impact their lives. Parents from both diagnosed and undiagnosed cohorts stated a belief that a diagnosis would not change their day-to-day lives.


*“How we go through day-to-day life, or the [medical] departments that Quinn interacts with, his environment, [a diagnosis] wouldn’t make a difference.”*
(Harry, father of Quinn, non-diagnostic result) 

Upon disclosure of the outcome of the exome testing, there was a divergence in the perception of results between the diagnosed and undiagnosed cohorts. Parents who received a diagnostic result described a greater shift in their everyday lives.

Parents who received non-diagnostic results described the genomic sequencing as another aspect of their child’s medical experience rather than something that would alter the future for their child or how they interacted with them.


*“I don’t think it really changed anything for me or the family...I don’t think that the genetic testing for her is the be all and end all.”*
(Jaqui, mother of Nina, non-diagnostic result) 

Although participants with a non-diagnostic result wanted clarity about their child’s future, they were adamant that a diagnosis would not change their child or how they viewed them.


*“Whatever label eventually gets put on, it won’t change who she is...getting a label isn’t going to change my feelings towards her or my care towards her. It will help with some understanding but the fact that we’ve had a knockback really hasn’t changed anything.”*
(Anna, mother of Zoe, non-diagnostic result) 

One aspect of the experience of parents whose child received a diagnosis was distinct. Some of these parents described a shift in the perception of their child, which enabled them to better understand and process the difficulties of their child’s condition.


*“For us, it’s very cathartic to understand that she is like she is, and this is why.” *
 (Kirsten, mother or Mia, diagnostic result)

A diagnosis also facilitated greater parental choice of how they spend their time as a family. Parents spoke of a shift away from the medicalization of their child to, instead, being able to focus on enjoying life with their child.


*“So you know that’s our option now right, because we have an answer. We can choose the time together to live the way he wants to live his life without having to sit there staring at each other going maybe [there is] something else that we need to do.”*
(Ellen mother of Roger, diagnostic result)


**Theme 3: Feelings of relief and disappointment**


Participants described the mixed emotions they felt regarding the results of the genomic testing. Both cohorts described the conflict between relief and frustration when considering the result received. Regardless of the outcome of testing, participants also described disappointment. For both groups of parents, this was related to lingering uncertainty. For parents who received non-diagnostic results, their disappointment came from not having answers.


*“I expected it. Look, I’m not going to lie, it gutted me...now I was like ‘that’s great, move along’. We’re never doing that again.” *
(Gina, mother of Riley, non-diagnostic result) 

Parents who did receive a diagnosis for their child explained how the diagnostic result had provided some answers and simultaneously raised further questions. This was due to the diagnosed conditions having only recently been published in literature. This was described as *“uncharted territory” (*Finn, father of Thomas, diagnostic result). Parents in this cohort shared how their excitement of receiving a diagnosis was mitigated by the scarcity of prognostic information and treatment options.


*“[The diagnosis was] exciting for us but at the same time, the fact that there was no information on it, that was hard. Getting the answer and then that side of it, like ‘oh there’s no information’, so not being able to understand it at that point in time...was quite tricky. So yeah, mixed emotions.”*
(Danielle, mother of Paul, diagnostic result) 

Together with feelings of disappointment, participants expressed relief at the results they received. Parents who received non-diagnostic results described feeling reassured that the testing had excluded many severe genetic conditions.


*“Having that foregone conclusion is quite hard, so having an exome you think, well they’ve ruled out a lot of things. They haven’t given me an answer but in some ways half of you is hoping not for an answer and half of you is hoping yes for an answer.”*
(Carly, mother of Winnie, non-diagnostic result) 

Parents who received diagnostic results also expressed relief but for different reasons. For some, this was because the diagnosis alleviated parental guilt surrounding the cause of the child’s condition.


*“A relief, just when you don’t have a diagnosis you always have this little element of guilt in the back of your mind.”*
 (Arabella, mother of Zander, diagnostic result)

Others spoke of how the diagnosis provided emotional relief, as it concluded the diagnostic odyssey and further investigations for their child.


*“For me personally, things have changed. I probably feel a lot more at ease and knowing that even though it [the diagnosis] took a long time, it’s been worth it in the long run. And to know that some of the testing now can stop is really relieving.”*
(Fiona, mother of Sam, diagnostic result) 


**Theme 4: Seeking connection**


Parents from both cohorts described the value of seeking connection with people whose lived experience mirrored their own. Even knowing that there were other families with the same experience was valued. For parents, acknowledgment that they were not alone helped alleviate their sense of isolation.


*“It’s a world that you’re in by yourself and to know that there’s other families that are similar, you go ‘ah ok, it’s not a battle that we’re facing alone.”*
(Isobel, mother of Penny, non-diagnostic result) 

Although parents from both groups were seeking connection with other families, the nature of the support they were searching for differed between the two groups. Parents who received non-diagnostic results described feeling isolated, expressing difficulty relating to parents of children with diagnosed conditions, as well as issues associated with accessing support services.


*“If you had that diagnosis, you’d meet other families, everyone could be on the same path. All of my friends have got kids with disabilities, they’ve all got diagnoses and they’ve all got their network.”*
(Flora, mother of Samantha, non-diagnostic result) 

The diagnosed cohort placed value on peer connection for different reasons. They emphasized the capacity of peer connection to provide prognostic information. As such, some actively sought connection with parents of older children with the same condition.


*“To see what the future might hold... from someone else’s experience.”*
(Bella, mother of Callum, diagnostic result) 

Families were reassured by learning of the positive experience of older children, often suggesting that they believed their child would follow the same positive trajectory.


*“We felt quite happy... because if we see someone with the same... problems grown up everything normal, then you feel happy you know. You see they’ve done well, they’re learning well, they lived a normal life you know.”*
 (Reyansh, father of Aarav, diagnostic result)


**Theme 5: Moving towards acceptance**


Participants spoke of developing acceptance; although this process of acceptance differed between participants, most acknowledged that it was something they needed to achieve as part of living with their child’s rare condition. Despite an ongoing sense of uncertainty about their child’s condition, many parents expressed hope for their child’s future. This was grounded in an expectation that advancements in genomic technology would lead to increased understanding in the future.


*“Genetics feels like a relief to me...thinking there’s something there that we might find out over the course of her lifetime. It’s not going to change anybody’s life but a little bit of relief.”*
(Diana, mother of Violet, non-diagnostic result) 

For many, the process involved finding a way to accept and live with not having a diagnosis for their child. For parents who received non-diagnostic results, acceptance was necessary to be able to move on after years of searching.


*“I think for my own sanity I’ve had to go with ‘we don’t know and we may never know’. We’ll just deal with whatever presents itself.”*
(Isobel, mother of Penny, non-diagnostic result) 

Parents whose children received a diagnosis shared how the result may not have provided the actionable information they sought but that they were able to use it to improve their outlook.


*“It wasn’t sweetness and light, and it isn’t, it didn’t change the prognosis, but it changed life, how our hearts work and that’s as important honestly.”*
(Ellen, mother of Roger, diagnostic result) 

## 4. Discussion

Our study explored the parental experience of participation in the UDP-Vic, finding that many parents across the undiagnosed and diagnosed cohorts described similar emotional responses to aspects of their participation in the program. Whereas the emotions experienced were similar, the cause of these emotions differed between the groups.

Parents were motivated to participate in the UDP-Vic by a desire for clarity and information about their child’s condition and care needs. Parents described the tumultuous nature of the diagnostic odyssey, with many hoping that participation in the UDP-Vic would bring an end to their diagnostic odyssey, although they harbored low expectations of this happening based upon prior experiences. Following genomic sequencing results, parents described the concurrent experience of relief and disappointment, which could be attributed to ongoing diagnostic or prognostic uncertainty. These findings are consistent with prior research [23,24,33]. It is important to emphasize that the participants were not split into individuals who experienced relief and individuals who experienced disappointment; rather, individuals could experience both emotions over the course of their journey or even both at the same time. Parents frustrated by the lack of a diagnosis expressed relief that particularly severe or life-limiting conditions had been excluded by comprehensive genomic sequencing. In their study about the long-term views of parents after genomic sequencing, Liang et al. [34] identified this relief as a potential misunderstanding of genomic information, as diagnostic genomic tests are limited in their ability to exclude genetic conditions. However, they also raised the possibility that the genomic test results have ‘emotional utility’ for parents. This appears to be consistent with our participants’ descriptions of relief and reinforces the idea that genomic tests have benefit beyond clinical utility. These mixed emotions are similarly reflected in studies conducted by Krabbenborg et al. [23], Rosell et al. [26] and Mollison et al. [35]. For parents whose child received a diagnosis, finally having a name for their child’s condition simultaneously ended the diagnostic odyssey and aided in relieving unresolved guilt. This was one of the notable differences between the two cohorts in our study and is also reflected in the findings of Liang et al. [34].

The possibility of disappointment, regardless of genomic sequencing results, is an important consideration for obtaining informed consent and pretest counseling. Bernhardt et al. [36] found that clinicians focused consent discussions on the uncertainty of non-diagnostic results and were concerned about patients’ misconceptions of non-diagnostic results. This was reflected by participants across both cohorts, who used terms such as, ‘diagnosis’, ‘an answer’ and ‘a name’ interchangeably when referring to a diagnosis. Delineating these terms with participants may be helpful, as often, a diagnosis may provide a name but not the answers parents are hoping for. We suggest that genomic sequencing consent processes should involve discussions around the notion that diagnostic results may have limited clinical utility but could still offer emotional utility.

The importance of parent-to-parent support groups throughout the diagnostic journey was emphasized by parents in both cohorts, suggesting that peer connection may relieve social isolation for parents of children with rare disease. Previous studies have demonstrated this [10,37,38]. The specific search of parents in the diagnosed cohort for families with the same condition in order to obtain prognostic information further highlights the desire for clarity and information felt by many families. Parents seemed reassured by a positive portrayal of their child’s condition, appearing hopeful that their child would follow the same health and wellbeing trajectory. There is likely benefit to exploring this in both the pretest and post-test counselling context in order to prepare parents for the often unique and unpredictable nature of rare diseases.

Parents described a sense of liberation through the perceived demedicalization of their child following genomic testing results. Parents whose child received a diagnosis were comforted by the idea that that they could shift away from searching and unnecessary investigations, with an increased capacity to spend time on other activities as a family. Fewer parents in the non-diagnostic cohort enjoyed a demedicalization of their child, attributing this to a sense that they had exhausted all available diagnostic routes. Nonetheless, many appeared to have reached a state of acceptance as a mechanism of coping. Still, many parents remained hopeful that more information would emerge about their child’s condition. This demonstration of hope and acceptance appeared to be foundational to parental wellbeing and coping. Studies by Mollison et al. [35] and Skinner et al. [39] similarly highlighted that genomic testing results allowed parents to shift away from searching while maintaining hope that future scientific advancements will yield treatments. Recent work by Liang et al. [34] demonstrated that parents do not regret choosing to undergo genomic sequencing for their child. Overall, participants in our study similarly reflected positively upon their decision to access genomic sequencing and partake in the UDP-Vic program.


**Practice Implications**


Our study highlights some of the unique issues that should be addressed in pretest counselling for genomic sequencing in the context of rare disease. Families being approached for genomic sequencing may benefit from clinicians highlighting the increased probability of a rare or novel gene finding when compared to traditional targeted genetic testing. Families should be made aware that such findings may not enable them to receive treatment or prognostic information, nor provide links to others with the same condition. However, it should be noted that genomic sequencing results may have emotional utility and provide an opportunity for the demedicalization of the day-to-day lives of families.


**Study limitations**


Participants in our study all provided consent to partake in the UDP-Vic and for their child to undergo genomic sequencing, thus excluding the perspective of those who declined to undergo genomic sequencing. However, we understand that such perspectives will be important going forward. This was also a retrospective study, so responses may have been impacted by participant recall bias.


**Directions for future study**


Rapid or ultra-rapid genomic sequencing is increasingly offered in the first instance for pediatric patients with suspected monogenic conditions, meaning fewer parents may experience a protracted diagnostic odyssey [13,14,40]. Parental experience of the ultra-rapid process may differ from that of traditional diagnostic journeys [41]. Additionally, parents could potentially have less time to bond with their child prior to diagnosis. Brett et al. [42] reported that most parents have no or mild decisional regret after participating in an ultra-rapid genomic sequencing program; however, the long-term psychological impact of a shortened diagnostic odyssey is yet to be explored.

## Figures and Tables

**Figure 1 jpm-12-01250-f001:**
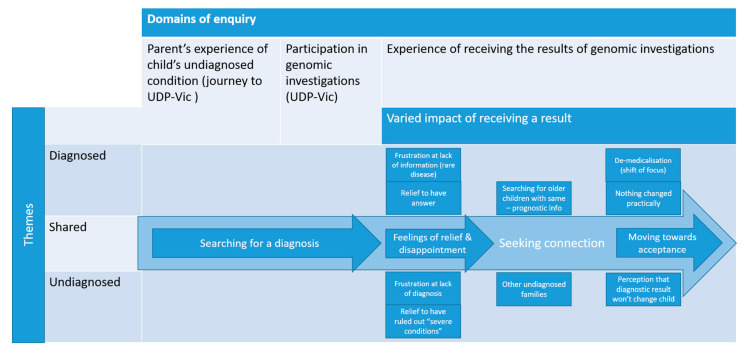
Comparison of parental experiences across both cohorts.

**Table 1 jpm-12-01250-t001:** Eligibility criteria used to assess the suitability of potential participants.

Eligibility Criteria	Exclusion Ctiteria
Capable of reading and understanding the informed consent document in English and consenting to participate in this study	Unable to provide informed consent
Over the age of 18	Unable to access a telephone or attend MCRI for interviews
Received a genetic result from UDP-Vic	Social, legal, and clinical issues identified by the treating VCGS clinicians that would make it inappropriate to contact the potential participant

**Table 2 jpm-12-01250-t002:** Participant information.

Participant Number	Classification of Genomic Result	Participant Pseudonym, Relationship to Child	Child Pseudonym	Age of Child at First Contact with Genetics as Reported by Participant	Age of Child at Diagnosis	Number of Individuals with Same Condition Globally, as Reported by Participant
P1	Non- diagnostic	Anna, Mother	Zoe, 22 years Female	At birth	-	-
P2	Diagnostic	Arabella, Mother	Zander 20 years Male	4 years	18 years	Not mentioned
P3	Diagnostic	Bella, Mother	Callum Male	4 months	10 years	10
P4	Diagnostic	Bree, Mother	Peter Died aged 5 years Male	At birth	4 years	2
P5	Non- diagnostic	Bridget, Mother	Yasmin 7 years Female	At birth	-	-
P6	Non- diagnostic	Carly, Mother	Winnie 8 years Female	2 or 3 years	-	-
P7	Diagnostic	Clara, Mother	Xavier 19 years Male	During pregnancy	18.5 years	16
P8	Diagnostic	Danielle, Mother	Paul 8 years Male	First weeks of life	Not known	Was 7th, diagnosed, now 33
P9	Non- diagnostic	Diana, Mother	Violet 26 years Female	2 years	-	-
P10	Diagnostic	Ellen, Mother	Roger 9 years Male	4 years	8 years	No one else in Australia
P11	Non- diagnostic	Ellie, Mother	Tara 5 years Female	6 months	-	-
P12	Diagnostic	Finn, Father	Thomas 7 years Male	3–6 months	7 years	4
P13	Diagnostic	Fiona, Mother	Sam 9 years Male	4 years	8 years	Approximately 250
P14	Non- diagnostic	Flora, Mother	Samantha 10 years Female	2 years	-	-
P15	Non- diagnostic	Gina, Mother	Riley 12 years Male	Prenatal period	-	-
P16	Non- diagnostic	Harry, Father	Quinn 3 years Male	First weeks of life	-	-
P17	Non- diagnostic	Isobel, Mother	Penny 18 years Female	3 years	-	-
P18	Non-diagnostic	Jaqui, Mother	Nina 7 years Female	2 months	-	-
P19	Diagnostic	Kirsten	Mia 11 years Female	2 or 3 years	10 years	10
P20	Non-diagnostic	Lana, Mother	Maddie and Nelly 14 years (twins) Female	9 years	-	-
P21	Diagnostic	Reynash, Father	Aarav 4 years Male	At birth	3 months	2

## Data Availability

No publicly available data were used in this study.

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
