# Peer review of "Receiving Genomic Sequencing Results through the Victorian Undiagnosed Disease Program: Exploring Parental Experiences"

_jpm, 2022, doi:10.3390/jpm12081250_

Round 1

Reviewer 1 Report

Congratulations on this interesting study, your paper is well-written and I am convinced that it will aid genetic counselors in their communication with families.

Although I understand your decision not to mention the diagnosis for the solved cases, as I suppose you anticipate possible identifiability, you might consider just mentioning the 10 etiologies, without assigning them to the individual comments. This might contribute to the significance of your report.

Author Response

Please see attached letter

Reviewer 2 Report

The Authors present an article that analyses the parents’ perspective regarding the impact of the receiving both diagnostic and non-diagnostic genomic sequencing results relatively to their children.

 I think that some aspects should be clarified by the Authors.

With regard to the Program UDP-Vic it would be better to describe how this program works not only in the abstract but also in the introduction and the difference of this program with other UDPs programs.

 The Authors considered only one parent for each children, but it is not clear if this is because both parents are not available or because the study is planned in this way. It would be interesting if the Authors would justify this aspect.

 The discussion should be implemented and better focused on the results as it maintains an overview with general statements rather than specific comments regarding the results obtained. The Authors discuss their results in comparison with other previous studies, however, they have to better highlight their results to giving originality and meaning to the interviews performed in their study. 

 References should be reviewed: many of the references are not reported as requested by the journal, for example all authors should be cited. 

Author Response

Please see letter attached

Round 2

Reviewer 2 Report

The Authors have answered to the reviewer comments and observations.